# SCI-YOLO11: An improved defect detection algorithm for transmission line insulators based on YOLO11

Junyan Wang[1], Yuqian Wang[2]*, Xun Li[1], Baoxi Yuan[3], Miaomiao Wang[4]

**1** Xijing University, Xi'an, Shaanxi, China, **2** Graduate Office, Xijing University, Xi'an, Shaanxi, China, **3** Xi'an Key Laboratory of High Precision Industrial Intelligent Vision Measurement Technology, Xijing University, Xi'an, Shaanxi, China, **4** North China Electric Power University, Baoding, China

* 936564267@qq.com

## Abstract

The detection of insulator defects in transmission lines is of paramount importance for the safe operation of power systems. However, small object detection faces numerous challenges, such as significant difficulty, substantial interference from complex backgrounds, and inconsistent annotation quality. These factors continue to constrain the performance of existing methods. To address these issues, this paper proposes an improved object detection algorithm named SCI-YOLO11, which optimizes the YOLO11 framework from three aspects: feature extraction, attention mechanism, and loss function. Specifically, to tackle the difficulties associated with small object detection, we replace conventional convolutions in the Backbone with SPDConv modules to enhance feature capture capabilities for small targets and low-resolution images while reducing computational overhead. To improve model accuracy further, we introduce the SE attention mechanism that adaptively adjusts the weights of feature channels to enhance the discriminative ability of insulator defect features. In response to the adverse effects caused by inconsistent annotation quality on defect image detection performance, we incorporate Wise-IoU-V3 loss function to optimize boundary box regression performance effectively mitigating negative impacts stemming from uneven annotation quality. Experimental results demonstrate that SCI-YOLO11 achieves a 3.2% improvement over baseline models in terms of MAP@0.5 metric; precision and recall rates increase by 2.6% and 3.7%, respectively. Additionally, its parameter count and floating-point operations decrease by 6% and 7.9%, respectively. These experimental findings substantiate the substantial improvements in detection accuracy, lightweight design, and robustness provided by SCI-YOLO11. This framework offers an effective technical solution for identifying defects in transmission line insulators.

**Data availability statement:** The dataset can be accessed at the following link: [https://www.kaggle.com/datasets/jmwangdeep/junyan-wang-insulator-dataset]. We hope that the availability of this dataset will contribute to research reproducibility and transparency. Please feel free to contact us if further clarification is needed. This dataset has been made publicly available on Kaggle (https://www.kaggle.com/datasets/jmwangdeep/junyan-wang-insulator-dataset) to support further research in related fields. The code required to replicate this study is available at: https://github.com/Xiaobaisci/SCI-yolo11.git.

**Funding:** The author(s) received no specific funding for this work.

**Competing interests:** The authors have declared that no competing interests exist.

## Introduction

The transmission lines are the core facilities that ensure the safe transmission of electricity within power systems, where insulators play a crucial role in providing electrical insulation and mechanical support. Insulators are among the most failure-prone components in electrical circuits. Their malfunction can disrupt the operation of the entire power system, potentially causing large-scale blackouts and significant economic losses [1]. Therefore, defect detection of insulators is of paramount importance during power inspections. Traditional inspection methods for transmission lines primarily rely on manual patrols or helicopter assistance; however, these approaches suffer from low efficiency, high costs, and considerable safety risks. In recent years, with advancements in drone technology, drone inspections have gradually emerged as an important means for power inspections due to their efficiency and safety features. Particularly when combined with computer vision techniques for automatic defect detection, drone inspections significantly enhance both inspection efficiency and accuracy [2]. Nevertheless, practical applications of drone inspections still face numerous challenges such as complex backgrounds, variations in lighting conditions, and limitations related to shooting angles.

Among these challenges, insulator defects, as small objects, are prone to feature loss in deep convolutional networks, resulting in missed or false detections. Additionally, when the background color closely resembles the color of the insulator, detection efficiency declines significantly. The rapid development of deep learning has provided new technological solutions to address these issues. For example, crime prediction research has successfully employed YOLOv8n for detecting small and intricate targets like weapons, achieving robust results even in complex scenarios [3]. These findings demonstrate the adaptability and effectiveness of deep learning models for small-object detection, providing valuable insights for insulator defect identification. Currently, target detection algorithms can be broadly categorized into single-stage detection algorithms and two-stage detection algorithms [4].

Single-stage target detection algorithms process images only once while simultaneously obtaining classification and location information about the targets. These algorithms operate at high speeds making them suitable for scenarios requiring real-time performance.Typical algorithms include YOLOv2 [5], YOLOv3 [6], and SSD [7], which are widely applied in real-time image processing scenarios involving drones due to their fast detection speed and low deployment costs.

Two-stage target detection algorithms-such as RCNN proposed by Girshick et al. [8], Fast RCNN [9], and Faster RCNN [10]-are typically referred to as region proposal-based methods for object detection. These algorithms achieve target identification through two phases: generating candidate regions during the first phase followed by precise classification and bounding box regression during the second phase. This approach demonstrates excellent performance against complex backgrounds; hence it is extensively utilized for tasks demanding high precision.

Despite each type having its advantages, both single-stage and two-stage target detection algorithms encounter three major technical challenges specific to insulator defect identification:

(1) Small Target Detection Challenge: Insulator defects are generally small-sized objects; existing object detection algorithms often struggle with effectively capturing detailed features when dealing with such small targets—especially within deep CNNs-resulting in missed detections. Although some optimization efforts have been made towards small-target recognition capabilities within certain models, fully capturing these minute details remains challenging due to scale differences among defective targets coupled with image resolution constraints.

(2) Complex Background Interference Issue: Images captured during insulator inspections frequently contain intricate backgrounds including wires trees along with weather variations that hinder effective recognition of targets. Such background interference tends to elevate both false positive rates (FPR) and missed detections (MDR), adversely affecting overall accuracy robustness across current algorithmic implementations. Enhancing model performance amidst complex backdrops while improving discrimination between foreground objects versus background elements continues being an urgent problem needing resolution.

(3) Balance Between Precision And Efficiency Problem: Drone inspection tasks impose stringent requirements regarding model responsiveness resource consumption levels. However, high-precision models usually entail substantial computational loads slower inference times. While improvements concerning precision have been achieved, excessive parameter counts computation demands restrict efficient operation on edge devices. Thus achieving optimal detectability whilst balancing accuracy computational efficacy represents a critical direction forward technologically speaking.

## Related work

Based on the aforementioned background issues, methods based on deep learning have gradually become a research hotspot for insulator defect detection in recent years. Compared to traditional methods, deep learning algorithms offer higher robustness and accuracy in complex backgrounds and small target detection scenarios. Additionally, they significantly enhance detection performance through feature extraction and optimization strategies. In the field of object detection, both single-stage and two-stage algorithms exhibit their strengths, providing different technical pathways for insulator defect detection tasks. Building upon this foundation, many scholars have conducted in-depth studies addressing the specific needs of insulator defect detection and proposed innovative improvement algorithms. Sadykova et al. [11] were the first to utilize YOLOv2 for real-time image data from drones to locate insulators; subsequently, they employed classifiers to assess the surface conditions of insulators to determine whether ice, snow, or water was present on their surfaces. In 2020, Ullah et al. [12] combined infrared thermal imaging technology with deep learning methods for analyzing insulator defects by extracting features using AlexNet and training classifiers to distinguish between defective and non-defective samples.

Chuanyang Liu et al. [13] enhanced YOLOv3 by adding multi-layer feature mapping modules and Feature Pyramid Networks (FPN), resulting in significant improvements in model speed and accuracy across various sizes of insulator detection tasks. Gujing Han et al. [14] proposed a lightweight detection algorithm based on Tiny-YOLOv4 [15], integrating self-attention mechanisms (Self-Attention [16]) with Efficient Channel Attention Network (ECA-Net) while reducing network complexity without compromising precision in detecting insulation damage within aerial images. In 2022, Han et al. [17] embedded the ECA-Net (Efficient Channel Attention Network) attention mechanism into the backbone network of YOLOv5. Additionally, they integrated a bidirectional feature pyramid network in the neck of the architecture, which preserved the original information of large-scale images. This approach significantly improved defect detection accuracy; however, there remains considerable room for enhancement in detecting broken string defects, and the model size is relatively large. Zhang et al. [18] proposed an enhanced version of YOLOv7 by replacing conventional convolutions in the backbone with depthwise separable convolutions,

resulting in improvements in both accuracy and training speed. Experiments were conducted on datasets for broken strings and flashover damage. In contrast to previous methods, He et al. [19] utilized YOLOv8 for further advancements and introduced a multi-insulator fault type detection algorithm called MFI-YOLO in 2024. This improvement fused GhostNet with multi-scale asymmetric convolution to construct C2f modules while replacing PANet with a residual-connected multi-scale feature fusion structure to achieve detection across four types: normal conditions, broken strings, damage, and flashovers. However, despite significant enhancements in mean Average Precision (mAP), the variety of defect types remains limited.

The application of two-stage detection algorithms has also made remarkable progress in insulator defect identification. For instance, Lin T et al. [20] conducted experiments using Faster R-CNN on images of insulators from different scenes to validate its usability and robustness across hundreds of images. Cheng H et al. [21], utilizing Faster R-CNN for recognizing insulators within aerial imagery achieved high-precision detections even amidst complex backgrounds.Despite the progress made in the field of insulator defect detection, existing research still faces challenges such as difficulties in detecting small targets, significant interference from complex backgrounds, and insufficient algorithm lightweighting. Therefore, further exploration of improved algorithms based on deep learning is crucial for enhancing the effectiveness of insulator defect detection. Drawing on the researchers' experiences and methodologies mentioned above, this paper proposes an improved single-stage object detection algorithm-SCI-YOLO11 based on YOLO11 [22], which has been specifically optimized for insulator defect detection tasks.

Compared to traditional methods, SCI-YOLO11 demonstrates significant advantages in terms of accuracy, speed, and resource consumption. The innovations of SCI-YOLO11 are primarily reflected in the following aspects:

(1) The quality of existing publicly available datasets is generally low, and the types of defects are relatively limited. For instance, the CPLID dataset [23] only includes images of insulators under clear weather conditions and features a narrow range of defect types, which fails to comprehensively reflect the characteristics of insulators in power line samples. Therefore, this paper contributes a high-quality dataset specifically designed for insulator defect detection. We captured images using DJI drones and enlisted experts from the electrical field for annotation, ensuring diversity and representativeness within the dataset. This dataset has been made publicly available on Kaggle (https://www.kaggle.com/datasets/jmwangdeep/junyan-wang-insulator-dataset) to support further research in related fields.

(2) In terms of model structure optimization, we have incorporated the SPDConv [24] convolution module into the basic framework. This module utilizes deformable convolution kernels to dynamically adapt to the irregular shapes of insulator defects (such as flashover or loss), effectively enhancing the model's ability to capture features of small targets. Furthermore, compared to traditional convolution modules (such as standard convolutions and conventional deformable convolutions), ordinary convolutions struggle with precise detection of small target defects due to their limited size on transmission lines. The SPDConv significantly reduces the number of model parameters while maintaining accuracy, making it more suitable for deployment on edge devices. However, the method proposed in this paper achieves a favorable balance between feature extraction and model complexity; it not only improves detection accuracy but also substantially decreases hardware resource consumption.

(3) Enhancement of Attention Mechanism: We integrated the SE attention mechanism into the backbone network, which adaptively adjusts the feature channel weights to effectively enhance feature representation capabilities. This integration significantly improves model detection accuracy, particularly in complex background scenarios. Compared to previous methods, such as Wei et al. [25] who employed CBAM, our approach more effectively mitigates interference from complex backgrounds while maintaining a constant model parameter count and improving small object detection precision. Furthermore, by incorporating the SE mechanism [26] into the backbone network compared to baseline models without an attention mechanism, this study enhances feature representation ability and boosts detection accuracy in challenging backgrounds.

(4) Improvement of Loss Function: We introduced the WIoU [27] loss function to optimize the bounding box regression process, effectively balancing training outcomes between high-quality and low-quality samples. This adjustment alleviates detection errors caused by uneven sample quality. The improvement notably enhances both robustness and practicality of the model. In contrast to the CIOU loss function used in baseline models, WIoU not only focuses on bounding box fitting accuracy but also emphasizes the significant contribution of low-quality samples to training results, thereby reducing performance degradation due to data annotation biases. Additionally, WIoU further accelerates convergence speed while enhancing model adaptability in complex scenes.

In summary, through these various improvements, we propose a novel SCI-YOLO11 that achieves lightweight design and real-time performance while maintaining high precision levels. This advancement contributes positively to defect detection technology for insulators during practical power inspections. The summary table of previous work is presented in Table 1.

## The SCI-YOLO11 algorithm

### The SCI-YOLO11 model

In response to the challenges of small target detection, significant background interference, and inconsistent annotation quality in the inspection of insulator defects on transmission lines, this paper proposes an improved YOLO object detection algorithm—SCI-YOLO11. The overall structure is illustrated in Fig 1. To address these issues, SCI-YOLO11 has undergone optimization in three key areas: feature extraction, attention mechanism, and loss function. Specifically:

(1) Optimization of Feature Extraction Module: The conventional convolutional branch within the backbone has been replaced with SPDConv modules to enhance the capability for capturing features from small targets and low-resolution images while simultaneously reducing computational load.

(2) Introduction of Attention Mechanism: The SE attention mechanism has been integrated into the backbone architecture, enabling the model to adaptively adjust channel weights. This enhancement improves both feature discrimination and representational capacity.

(3) Loss Function Optimization: To mitigate the impact of inconsistent annotation quality on detection performance, we have designed a Wise-IoU-V3 loss function that allows the detector to better balance high- and low-quality anchor boxes during boundary box regression processes, thereby enhancing overall detection efficacy.

Through these optimizations, SCI-YOLO11 can focus more effectively on critical regions within input images' feature information. It demonstrates exceptional performance in detecting small targets amidst complex backgrounds while significantly improving accuracy and robustness in detection outcomes.

**Table 1. Summary of previous work.**

| Reference | Methodology and Key Features | Limitations |
|---|---|---|
| [11] | YOLOv2 + Classifier, UAV, surface condition analysis | Limited small object detection |
| [12] | AlexNet+Classifier, infrared imaging | Binary classification only |
| [13] | YOLOv3 + FPN, improved speed & accuracy | High model complexity |
| [14 and 17] | YOLO+ECA-Net, lightweight, self-attention | Weak small object detection |
| [18,19] | YOLOv8 + GhostNet, multi-scale fusion, multi-fault types | Limited defect diversity |
| [20,21] | Faster R-CNN, multi-scene robustness | High computational cost |
| OURS | lightweight design, SPDConv, SE, WIoU | Needs testing in diverse environments |

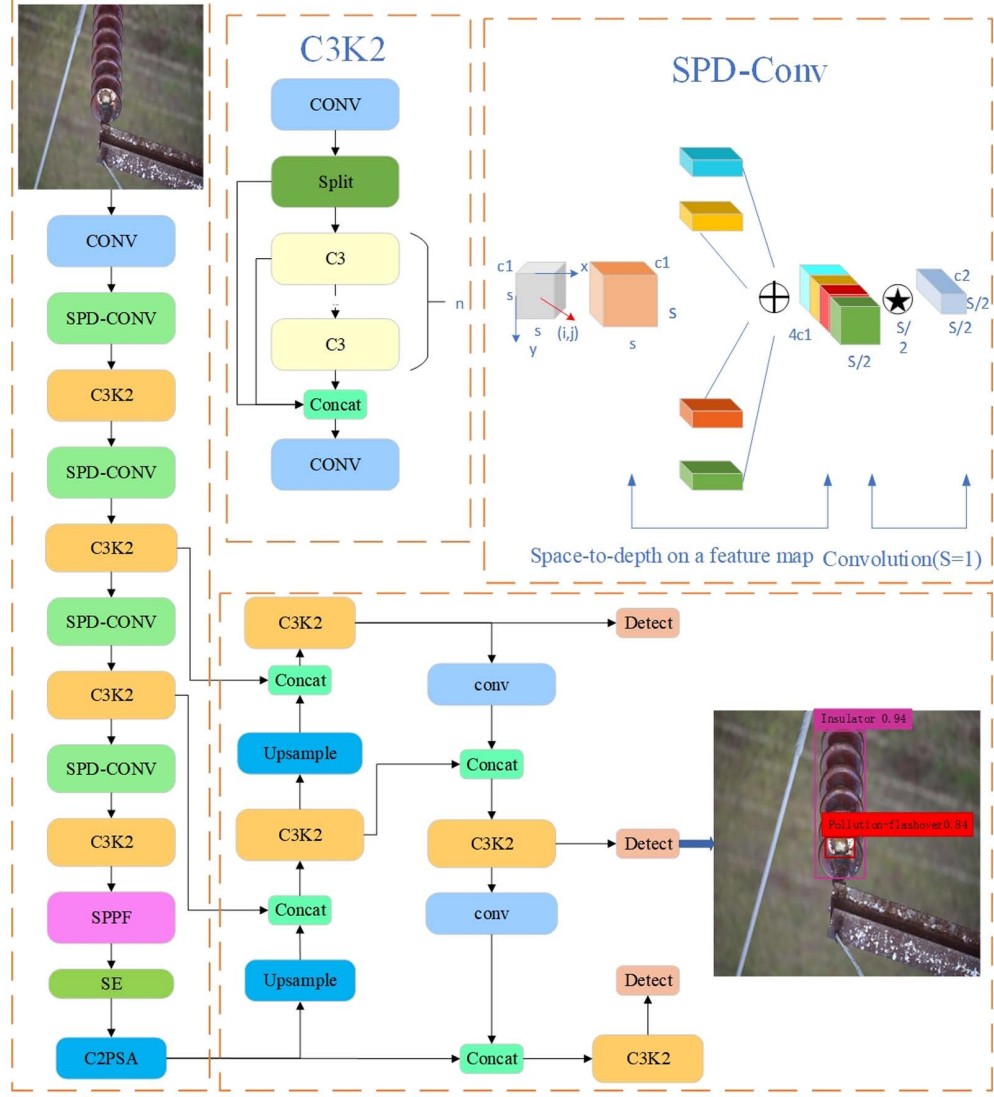

**Fig 1. Structure of the SCI-YOLO11 network.**

### Spatial depth transformation convolution

To address the performance degradation of traditional Convolutional Neural Networks (CNNs) in low-resolution image and small object detection tasks, this paper introduces the SPD-Conv (Spatial-to-Depth Convolution) module into the network. This module replaces strided convolutions and pooling layers, thereby reducing the loss of fine-grained information and enhancing the capability to capture features from small objects and low-resolution images. The SPD-Conv (Spatial-to-Depth Convolution) module is designed to enhance the feature extraction capabilities of low-resolution images and small targets by integrating spatial information into the channel dimension. The specific implementation process is shown in Table 2. It primarily consists of the following two steps:

(1) Space-to-depth transformation:

Given an input feature map X with dimensions $S \times S \times C_1$, the Space-to-Depth (SPD) operation divides the feature map into $scale^2$ sub-regions $f_{i,j}$, where each sub-region is defined as show in Eq 1:

$$f_{i,j} = X[i : S : sacle, j : S : scale, :]$$
(1)

Here, i, $j \in \{0,1,...,scale-1\}$. Each sub-region has dimensions $S/scale \times S/scale \times C_1$. These sub-regions are concatenated along the channel dimension to generate a new feature map X', with dimensions as show in Eq 2:

$$X'(\frac{S}{scale}, \frac{S}{scale}, C_1 \times scale^2)$$
(2)

This transformation reduces the spatial resolution while increasing the channel depth, effectively preserving rich spatial information.

(2)   Non-strided convolution

The transformed feature map X' is subsequently passed through a non-strided convolutional layer to extract enhanced features. The convolution operation is defined as show in Eq 3:

$$Y = X' * W + b$$
(3)

Where W and b represent the convolutional kernel weights and bias terms, respectively. This process captures high-level semantic information and further optimizes feature representation. The final output feature map Y has dimensions: Y(s/scale,s/scale,$C_2$), Here, $C_2$ represents the number of convolutional kernels. In this study, the SPD-Conv module replaces the standard convolution in YOLO, enhancing the model's performance in detecting small objects and low-resolution targets. By incorporating the space-to-depth mechanism, the module captures finer-grained details, thereby improving the model's ability to detect small targets in complex backgrounds. The operational steps for scale = 2 are illustrated in Fig 2.

## Attention mechanism: SE attention

In order to enhance the network's ability to capture features of small objects and critical regions, this paper introduces the Squeeze-and-Excitation (SE) attention mechanism into the backbone of SCI-YOLO11. This mechanism adaptively adjusts

**Table 2. Algorithm operation steps.**

| Step | Description |
|---|---|
| input | Feature map $X$ of size $(S \times S \times C_1)$<br>Downsampling factor $Scale$<br>Convolution kernel $W$<br>Output channel size $C_2$ |
| 1 | Split feature map $X$ into $Scale^2$ sub-feature maps:<br>$f_{i,j} = X[i : S : sacle, j : S : scale, :]$ |
| 2 | Concatenate all sub-feature maps along the channel dimension:<br>$X' = Concatenate(f_{0,0,}, f_{0,1}..., f_{scale-1,scale-1})$<br>The concatenated $X'$ has size $(\frac{S}{scale}, \frac{S}{scale}, C_1 \times scale^2)$ |
| 3 | Apply non-strided convolution to $X'$:<br>$Y = Conv(X', W)$<br>The output feature map $Y$ has size $(\frac{S}{scale}, \frac{S}{scale}, C_2)$ |
| Output | feature map $Y$ of size $(\frac{S}{scale}, \frac{S}{scale}, C_2)$ |

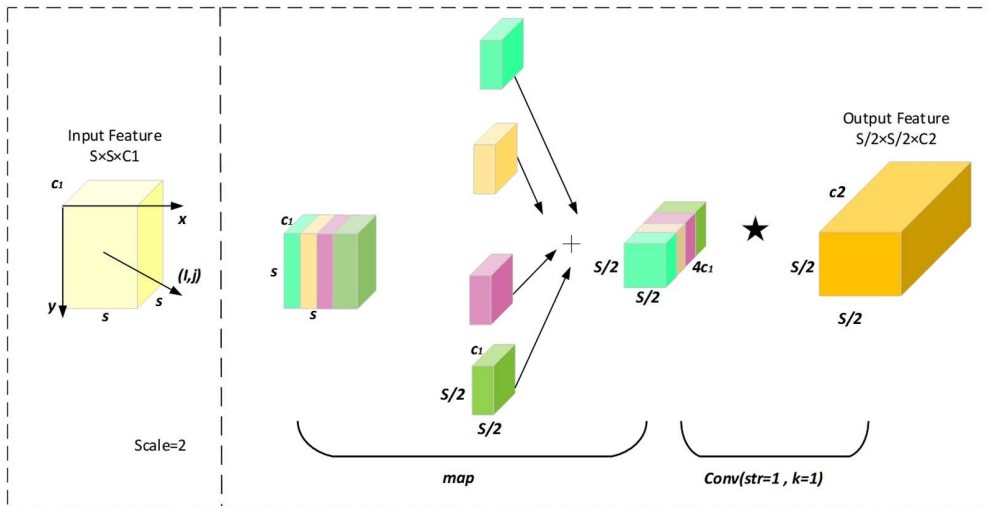

**Fig 2. Structure diagram of SPDConv.**

the weights of feature channels, enabling the network to focus more on features that contribute to detection tasks while suppressing irrelevant or distracting features, thereby improving detection performance.

The core components of the SE mechanism consist of two steps: Squeeze and Excitation (as illustrated in Fig 3). First, global average pooling is employed to compress spatial information from input features into a global feature vector, extracting global information for each channel. Subsequently, a fully connected network along with a nonlinear activation function is utilized to adaptively adjust the weights for each channel. These adjusted weights are then applied for recalibrating the original feature map, emphasizing important channel representations.

In SCI-YOLO11, the introduction of the SE mechanism significantly enhances its ability to express features related to small target insulators and complex scenes, thereby improving detection accuracy and robustness. Experimental results demonstrate that this improvement leads to notable enhancements across various metrics such as mean Average Precision (MAP), recall rate, and precision rate, validating its effectiveness in small object detection and background interference suppression.

In order to enhance the network's ability to capture features of small targets and critical regions, this paper introduces the Squeeze-and-Excitation (SE) attention mechanism between the SPPF layer and C2PSA layer in the backbone of

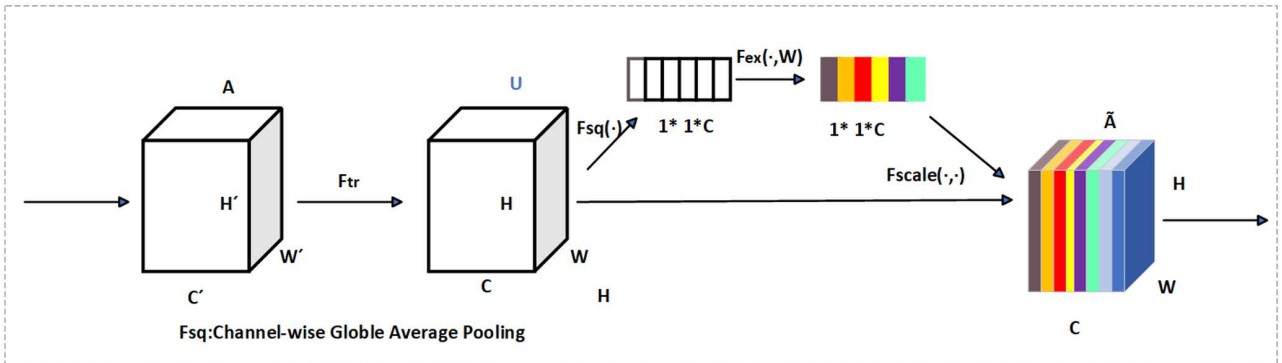

**Fig 3. Structure diagram of the SE attention mechanism.**

SCI-YOLO11. Global average pooling is performed on the input feature map $F_{in} \in \mathbb{R}^{C \times H \times W}$ to obtain a global feature vector along the channel dimension as shown in Eq 4:

$$G = GAP(F_{in}) \Rightarrow G_c = \frac{1}{H \cdot W} \sum_{i=1}^{H} \sum_{j=1}^{W} F_{in}(c, i, j), c \in [1, C] \tag{4}$$

The operation extracts global semantic information from each channel by compressing the spatial dimensions. Subsequently, the global feature G is input into a lightweight network for nonlinear transformation: $Z = \sigma(Relu(W_2 \cdot \tanh(W_1 \cdot G)))$, Here, $W_1 \in \mathbb{R}^{\left(\frac{C}{16}\right) \times C}$, $W_2 \in \mathbb{R}^{\left(\frac{C}{16}\right) \times C}$ are learnable parameter matrices, $\tanh(\cdot)$ and $\sigma(\cdot)$ represent the tanh and sigmoid activation functions, respectively. The final output is to generate a weighted feature map by performing element-by-element multiplication, as shown in Eq 5.

$$F_{out} = F_{in} \cdot Z \tag{5}$$

## Loss functions

The overall loss function of YOLO11 consists of two components: classification loss and regression loss. The classification loss ($L_{cls}$) is calculated using binary cross-entropy loss (BCE), while the regression loss incorporates both Complete Intersection over Union Loss (CIoU) [26] and Distributed Focal Loss (DFL). CIoU is employed to assess the intersection over union loss between the predicted bounding box and the target bounding box ($L_{box}$), whereas DFL measures the distance loss between the regression predicted bounding box and the target bounding box ($L_{dfl}$). Therefore, the total loss function of YOLO11 is denoted as LLoss, as shown in Eq 6.

$$L_{LOSS} = \lambda_1 L_{CLS} + \lambda_2 L_{box} + \lambda_3 L_{dfl} \tag{6}$$

In the equation, $\lambda_1 \lambda_2$, and $\lambda_3$ represent the weight values.

Due to the fact that the CIoU loss function only considers geometric factors such as the distance between predicted boxes and ground truth, as well as aspect ratios, it lacks attention to the quality of annotated samples. This limitation can lead to slower convergence rates and poorer generalization performance. Despite existing work assuming that the training data consists of high-quality samples and focusing on enhancing the fitting capability of boundary box regression loss, blind optimization of low-quality samples may compromise localization accuracy. Although Focal-EIoU v1 was proposed to address this issue, its static focusing mechanism (FM) fails to fully leverage the potential of a dynamic non-monotonic FM—this mechanism can adjust gradient allocation based on training dynamics [28]. Based on this observation, we introduce a distance attention mechanism built upon the original Wise-IoU framework, proposing Wise-IoU-V3. (A schematic representation of this regression process is illustrated in Fig 4). This approach aims to achieve precise gradient prioritization for high-quality anchor boxes while mitigating the influence of low-quality samples.

The design objective is as follows: by dynamically adjusting gradient weights, we prioritize optimizing high-quality anchor boxes and suppressing the impact of low-quality samples.

(1) Design of dynamic non-monotonic focusing mechanism

Traditional loss functions (such as Focal-EIoU) employ a static non-monotonic focusing strategy that cannot adapt to dynamic changes during training phases. To address this limitation, we introduce the degree of outlier (β), as shown in Eq 7, which quantifies the deviation between the current anchor box loss and the global average loss:

$$\beta = L_{IOU}^* / \overline{L}_{IOU} \tag{7}$$

The term "$L^{*}_{IOU}$" refers to the IoU loss of the current anchor box, while "$\overline{L}_{IOU}$" denotes the global average loss, which is updated through exponential moving averages. Based on "β" we define a dynamic focusing coefficient as follows: $r=β/(α^{β-α})$. when β=δ, r = 1 represents the optimal gradient allocation point (maximum gradient gain).

**Non-monotonicity:** When β > δ (low-quality samples), r approaches 0, suppressing the gradient,when β < δ (high-quality samples), γ increases linearly with β, prioritizing optimization.

The hyperparameters α and δ control the sensitivity and dynamic threshold of anomaly degree. Parameter δ is dynamically updated using momentum m as shown in Eq 8:

$$\delta_{t+1}=m \cdot \delta_t+(1-m) \cdot \overline{L_{IOU}} \tag{8}$$

At the initial stage of training, $\overline{L_{IOU}} \approx 1$ and m = 0.1, which allows the model to focus on high-confidence regions for rapid convergence. In the later stages, as m increases, δ gradually stabilizes to the overall average value, thereby balancing ordinary samples and low-quality samples.

(2) Introduction of distance-aware attention weighting:

To enhance sensitivity to local regions with significant errors, a distance penalty term $R_{WIOU}$ is defined, as shown in Eq 9

$$R_{WIOU}=exp\left[\frac{(x-x_{gt})^2+(y-y_{gt})^2}{\left(W_g^2+H_g^2\right)^{*}}\right] \tag{9}$$

Here, $(W_g, H_g)$ represent the width and height of the bounding box for the ground truth, while $*$ denotes the detached calculation to avoid gradient instability. This weighting mechanism penalizes predictions that deviate significantly from the ground truth, encouraging the model to prioritize optimization in regions close to the target. The final loss function formula is shown in Eq 10:

$$L_{Wise-IOUv3}= r \cdot L_{WiouV1}, \; L_{IOUv1}=R_{WIOU} \cdot L_{IOU} \tag{10}$$

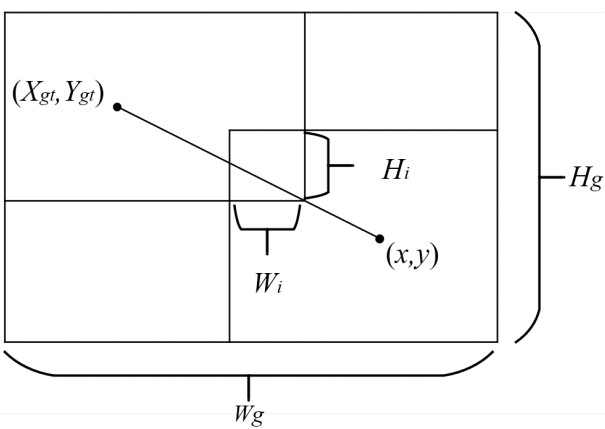

**Fig 4. Wise-IOU regression diagram.**

## Experimental results and analysis

### Experimental environment and parameter processing

(1) The following experiments were conducted on an NVIDIA GeForce RTX 4090D (24GB) GPU and an Intel(R) Core(TM)i9-14900K CPU. The deep learning framework utilized in this study is PyTorch 2.1.1, and the programming language employed is Python 3.10.

(2) In this experiment, stochastic gradient descent (SGD) was used to optimize the network parameters. The initial learning rate was set to 0.01, with a momentum of 0.937 for the learning rate and a weight decay of 0.0005. A batch size of 64 was applied, with input images sized at 640 pixels by 640 pixels, and the number of epochs was set to 300.

### Dataset preparation and processing

This study utilizes a dataset collected along the transmission lines in Northwest China, primarily through data acquisition via drones equipped with optical cameras. The collection equipment consists of the DJI Matrice 300 RTK drone and the Zenmuse Z30 gimbal camera [29]. This drone is characterized by its extended flight time and high stability, while the Zenmuse Z30 supports 30x optical zoom, enabling clear capture of detailed features of distant insulators. By strategically planning flight paths, we covered various operational conditions and perspectives, thereby constructing a diverse and high-quality dataset that provides foundational support for research on defect detection in transmission line insulators. Fig 5 depicts the detailed process of image acquisition utilizing drones. In our experiments, we selected two types of insulators: glass insulators (Two High Glass) and polymer insulators (Insulator). For these two categories of insulators, we identified five distinct types of faults: glass dirtiness (Glass Dirty), glass damage (Glass Loss), polymer dirtiness (Polymer Dirty), polymer damage (Broken Disc), and insulation flashover (Pollution Flash). We performed super-resolution reconstruction on the original dataset; after optimization, it comprised a total of 2,828 images. The dataset was randomly divided into training set with 2,150 images, test set with 287 images, and validation set with 391 images at a ratio of 7:2:1. Additionally, we employed the LabelImg algorithm tool to annotate the images; part of the completed dataset is shown in Fig 6.

### Evaluation metrics

The present study employs five evaluation metrics to assess the performance of the model, namely Mean Average Precision (MAP), Precision (P), Recall (R), Parameters, and Computational Complexity (FLOPs).

In object detection, Mean Average Precision is commonly utilized to evaluate model performance. The formula is shown in Eqs 11,12:

$$MAP = \frac{\sum_{i=0}^{n} AP(i)}{n} \tag{11}$$

$$AP = \int_{0}^{1} P(R)dR \tag{12}$$

The variable n represents the number of image categories, with this study setting n = 7. The variable i denotes the number of detection attempts, and AP refers to the average precision for a single category. In this paper, we employ mAP@0.5, where IoU is set at 0.5. We calculate the AP for each image category and then take the average across all categories.

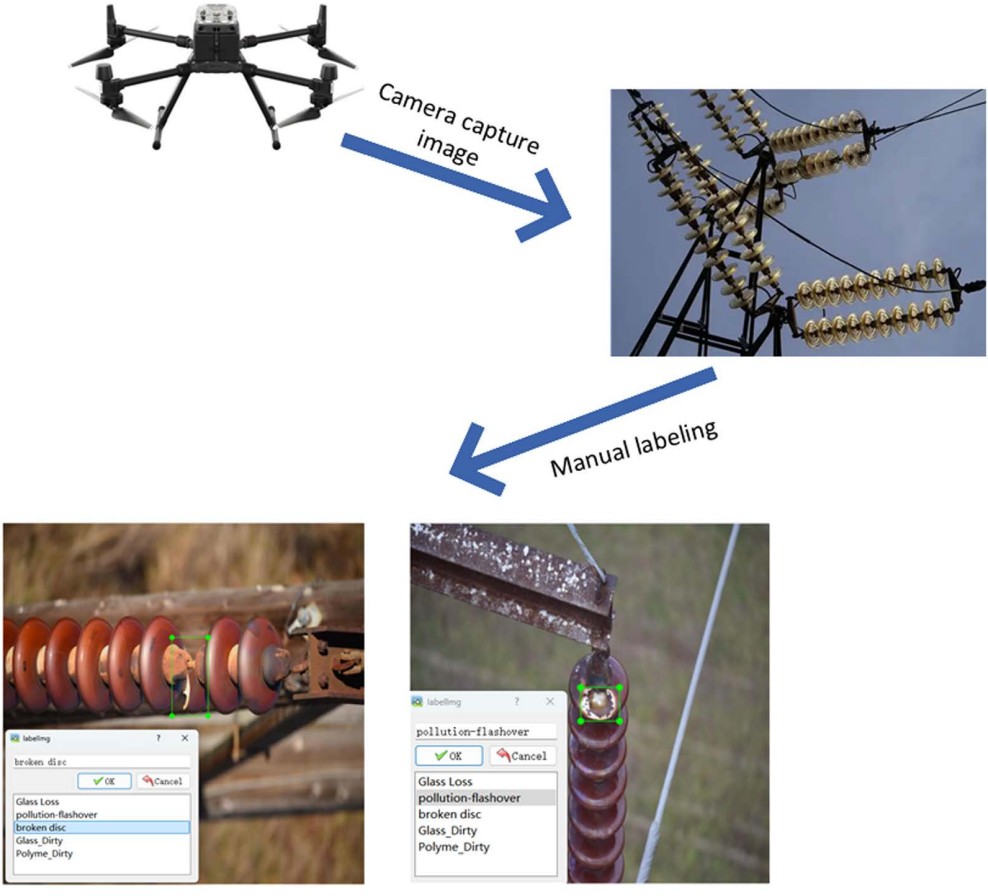

**Fig 5. The process of data collection and annotation by drones.**

Precision indicates the accuracy of a model's detection, specifically the proportion of true positive samples among those classified as positive. Recall reflects the model's ability to identify actual positive instances, representing the ratio of correctly identified positives out of all true positives. The formula for these two indicators is shown in Eqs 13,14:

$$Precision = \frac{TP}{TP + FP} \tag{13}$$

$$Recall = \frac{TP}{TP + FN} \tag{14}$$

The term TP (True Positive) refers to the instances where the model successfully predicts positive cases. FP (False Positive) denotes negative instances that are incorrectly classified as positive by the model, while FN (False Negative) indicates positive cases that are misclassified as negative. FLOPs represent the computational complexity of a model, quantifying the number of floating-point operations involved during its execution, measured in Gflops (G). The parameter count (Params) signifies the total number of trainable weights and biases within the model, which is commonly used to assess both the complexity and storage cost associated with it.

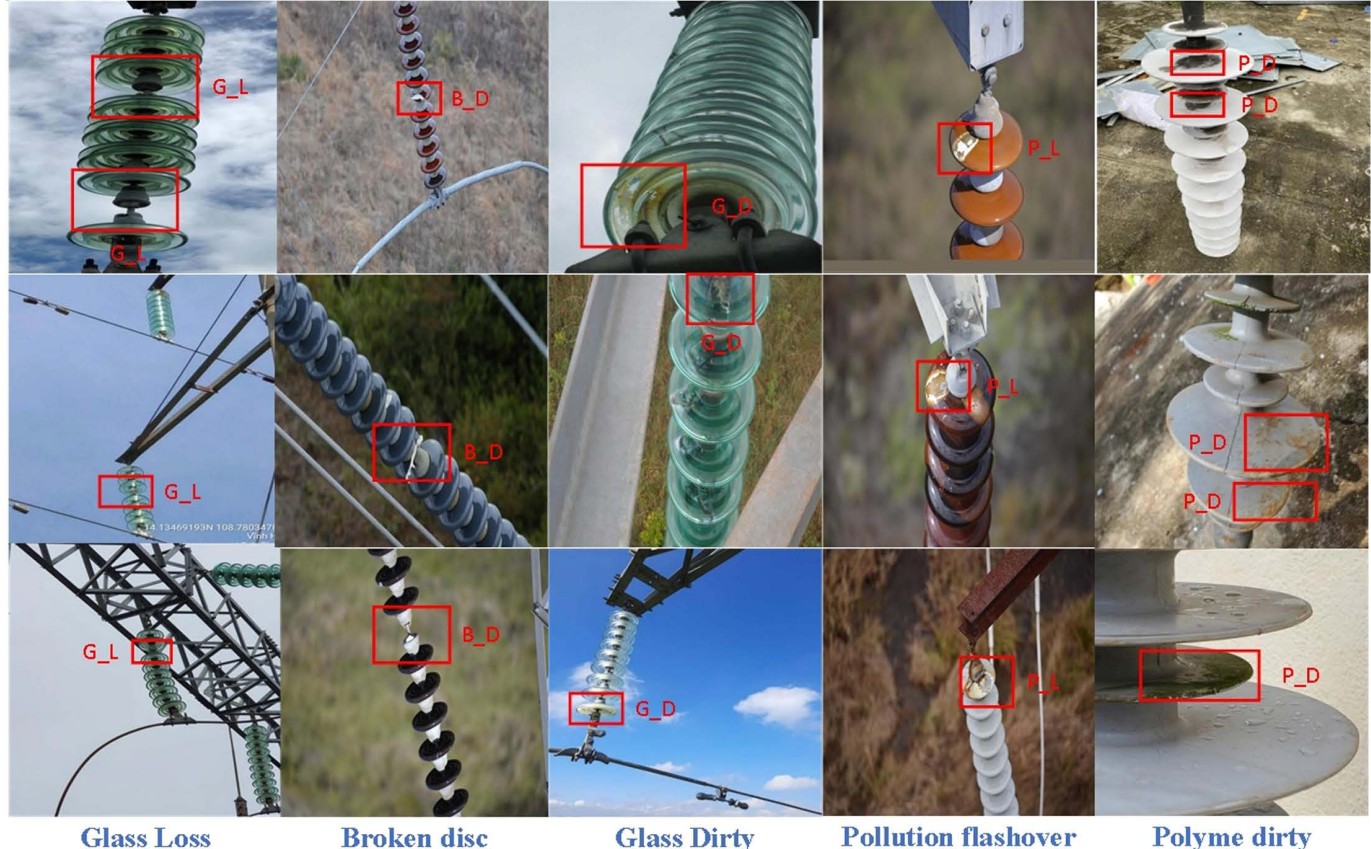

**Fig 6. Dataset for insulator defect detection.**

## Experimental results and analysis

**Ablation experiments.** In order to validate the contributions of various modules to the performance of the SCI-YOLO11 algorithm, this study conducted ablation experiments that sequentially assessed the effects of the SPDConv module, SE attention mechanism, and Wise-IoU-V3 loss function. The experimental results are presented in Table 3.

In terms of optimizing the feature extraction module, the introduction of the SPDConv module alone resulted in an increase in mAP@0.5 from 0.862 to 0.872, while simultaneously leading to a significant reduction in both parameter count and FLOPs. This indicates that SPDConv not only enhances the capability to capture features of small objects but also reduces the computational complexity of the model. When incorporating the SE attention mechanism on this basis, mAP@0.5 further improved to 0.878, with concurrent increases in Precision and Recall, thereby validating the substantial advantages of the SE mechanism in enhancing feature representation discrimination.

Regarding multi-module collaborative optimization, models that integrate both SPDConv and SE modules maintain high accuracy while still achieving a low parameter count and FLOPs, thus realizing lightweight and efficient modeling. Finally, after combining with the Wise-IoU-V3 loss function, mAP@0.5 increased further to 0.894—reaching its highest value across all experiments—while also optimizing boundary box regression performance effectively alleviating issues related to uneven annotation quality.

As illustrated in Fig 7, the performance differences between the baseline model and SCI-YOLO11 are analyzed in terms of mAP@0.5 and F1 score. Specifically, Fig 7A presents the Precision-Recall curve for the baseline model,

**Table 3. Results of ablation experiments.**

| Model | SPDConv | SE | Wise-iou | Map@0.5 | P | R | Flops | Params |
|-------|---------|-----|----------|---------|-------|-------|-------|--------|
| Yolo11 | × | × | × | 0.862 | 0.863 | 0.818 | 6.3 | 2.56 |
| | √ | × | × | 0.872 | 0.865 | 0.846 | 5.5 | 2.3 |
| | × | √ | × | 0.878 | 0.876 | 0.832 | 6.4 | 2.7 |
| | √ | √ | × | 0.877 | 0.899 | 0.834 | 5.8 | 2.4 |
| | √ | √ | √ | 0.894 | 0.889 | 0.855 | 5.8 | 2.4 |

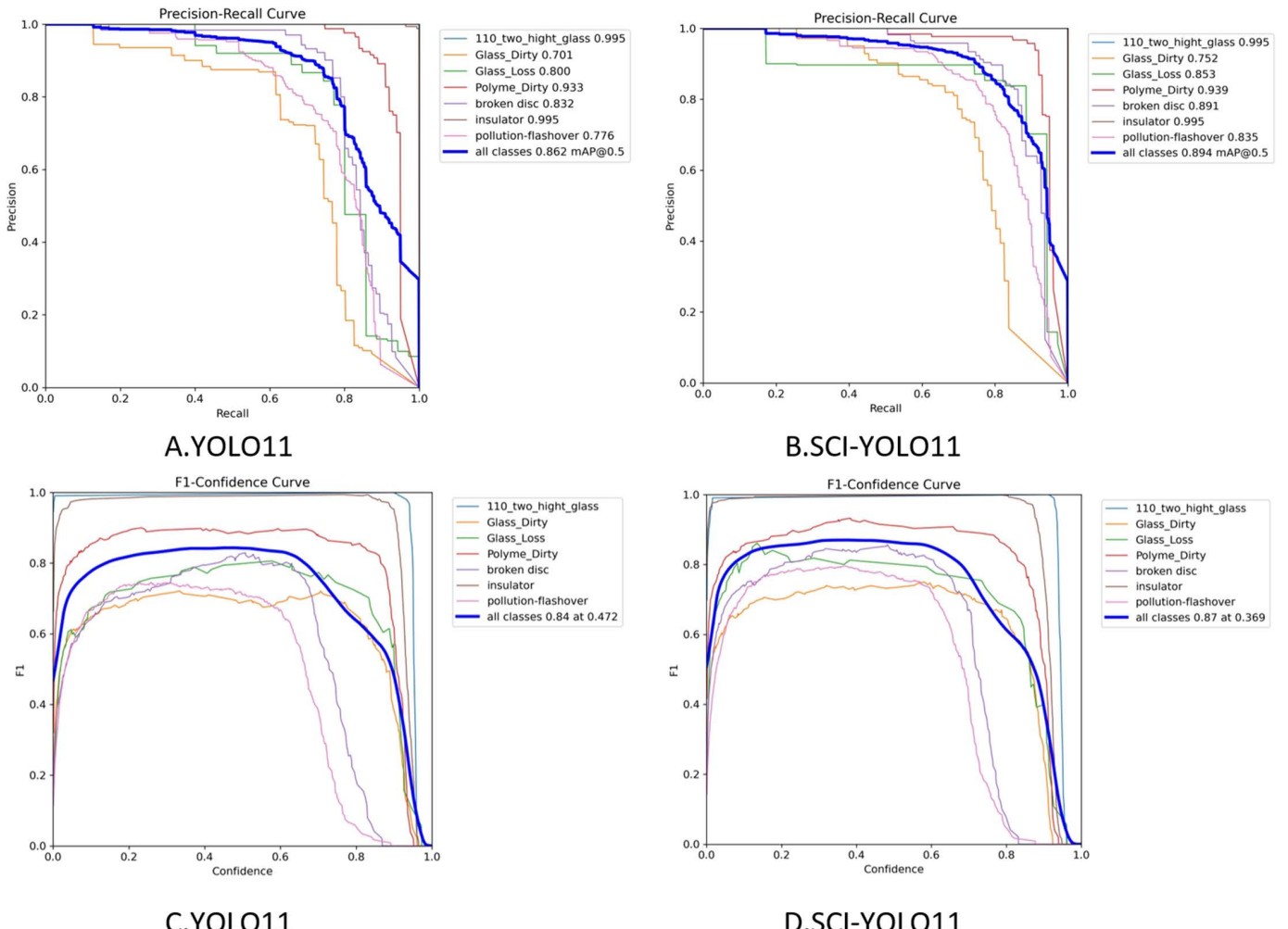

**Fig 7. Comparison of effects before and after improvement.**

achieving an mAP@0.5 of 0.862, while Fig 7B illustrates the Precision-Recall curve for SCI-YOLO11, which improves the mAP@0.5 to 0.894, reflecting a 3.2% enhancement. Regarding the F1 score, Fig 7C shows the F1-confidence curve of the baseline model with a peak F1 score of 0.84, whereas Fig 7D demonstrates the F1-confidence curve for SCI-YOLO11, where the F1 score rises to 0.87. These results underscore the effectiveness of the proposed improvements, including the incorporation of SPDConv, SE attention mechanism, and Wise-IoU-V3 loss function. Together, these enhancements

significantly improve the model's ability to detect small objects and mitigate the influence of complex backgrounds, leading to notable gains in detection accuracy and robustness. Furthermore, ablation studies confirm that these modifications not only boost detection performance but also contribute to reducing the computational complexity of SCI-YOLO11, demonstrating its practical applicability for real-world scenarios.

**Visualization of ablation experiment results through heatmaps.** In order to intuitively demonstrate the enhancement effects of various improvement strategies on model performance, this study employs heatmaps to visually compare the feature response characteristics of SCI-YOLO11 and a baseline model during the detection process.

Heatmaps effectively illustrate the degree of attention and activation strength that the model allocates to different target areas, thereby facilitating a clear reflection of performance differences in feature extraction, object recognition, and bounding box regression. Through this visualization approach, we can more readily perceive the actual contributions of each improvement strategy to object detection efficacy. This provides explicit direction and justification for subsequent model optimization and design efforts.

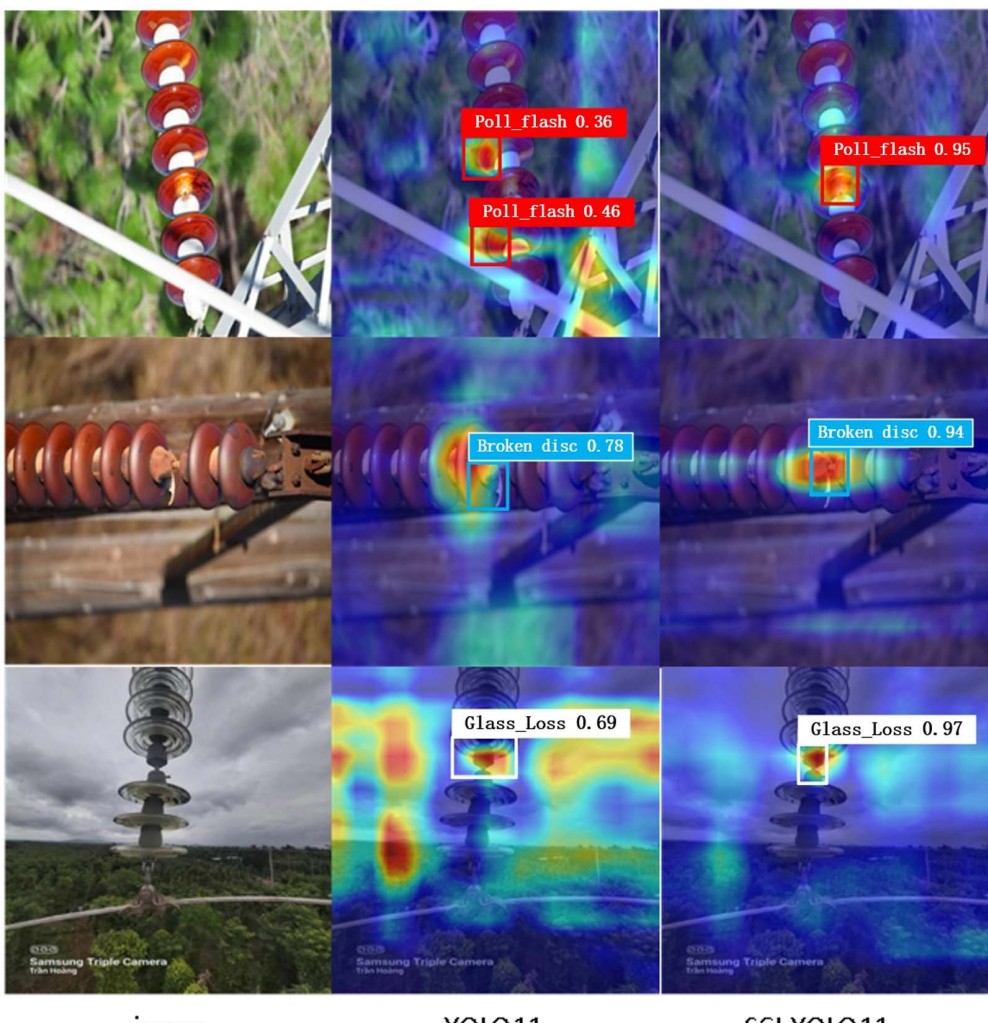

**Fig 8. Visualization of heatmaps before and after improvement.**

In the heatmap visualization of the ablation experiments (as shown in Fig 8), a clear observation can be made regarding the performance differences of the model before and after the introduction of key improvement strategies. Compared to the baseline model, SCI-YOLO11 exhibits significantly enhanced feature responses in target areas, with a marked reduction in both false positives and missed detections, particularly demonstrating outstanding performance in scenarios involving small objects and complex backgrounds. Specifically, the incorporation of SPDConv has strengthened the capability for detailed feature representation, while the SE attention mechanism further optimizes feature weight distribution, resulting in more concentrated and clearer high-response regions within target areas. The application of Wise-IoU-V3 loss function effectively alleviates issues related to inaccurate bounding box fitting, leading to a significant increase in prediction accuracy. Overall, the results from heatmap visualization intuitively illustrate the practical effects of each improvement, thereby further validating SCI-YOLO's effectiveness in enhancing precision and optimizing model performance.

**Comparative experiments.** The performance of various object detection models in the task of detecting defects in insulators along transmission lines is compared in Table 4, including metrics such as Mean Average Precision (mAP@0.5), Precision (P), Recall (R), and model complexity (FLOPs and parameter count). The improved model (OURS) demonstrates significant advantages across all evaluated metrics.

In terms of mAP@0.5, the OURS model achieved a score of 0.894, representing an improvement of 4.8% over Faster R-CNN (0.853), 4.1% over SSD (0.859), and 3.4% over YOLOv8 (0.865). For Recall (R), the OURS model reached a value of 0.855, which is an increase of 4.1% compared to both Faster R-CNN (0.821) and YOLOv8 (0.821). This further substantiates its exceptional performance in complex backgrounds and small object detection tasks.

Moreover, the OURS model excels in lightweight design with a parameter count of only 2.4M, which represents a reduction of 94.1% compared to Faster R-CNN's parameters at 41M and a decrease of 90.9% relative to SSD's parameters at 26.3M. In terms of FLOPs, OURS requires merely 5.8G—significantly lower than Faster R-CNN's requirement of 180G and SSD's requirement of 99.6G, resulting in reductions of approximately 96.8% and 94.2%, respectively.

When compared to other lightweight models from the YOLO series—such as YOLOv8 with its FLOPs at 8.0G and YOLO11 at approximately 6.3G—the OURS model shows reductions of about 28.4% and 7.9% while achieving superior performance without compromising on low complexity.

To further validate the model's performance, Fig 9 presents the variation curves of MAP@0.5 throughout the training process for various models. It is evident that the OURS model (represented by the black curve) exhibits a rapid convergence rate in the early stages of training, followed by a steady improvement that consistently outpaces other models. More importantly, the curve for the OURS model demonstrates minimal fluctuations throughout training, indicating a high degree of stability. This suggests that its optimization strategy is effective in both feature extraction and loss function design. Such robustness is particularly critical for practical applications, as it implies that the model not only performs well

**Table 4. Comparative experimental results.**

| Model | Map@0.5 | P | R | Flops (G) | Params (M) |
|---|---|---|---|---|---|
| Faster-RCNN | 0.853 | 0.876 | 0.821 | 180 | 41 |
| SSD | 0.859 | 0.601 | 0.838 | 99.6 | 26.3 |
| YOLOV5 | 0.863 | 0.873 | 0.812 | 7.1 | 2.5 |
| YOLOV6 | 0.856 | 0.865 | 0.794 | 11.8 | 4.2 |
| YOLOV8 | 0.865 | 0.892 | 0.821 | 8.1 | 3.0 |
| YOLOV9t | 0.858 | **0.904** | 0.825 | 7.6 | 1.9 |
| YOLOV10 | 0.866 | 0.82 | 0.807 | 8.2 | 2.6 |
| YOLO11 | 0.862 | 0.88 | 0.818 | 6.3 | 2.56 |
| OURS | **0.894** | 0.889 | **0.855** | **5.8** | 2.4 |

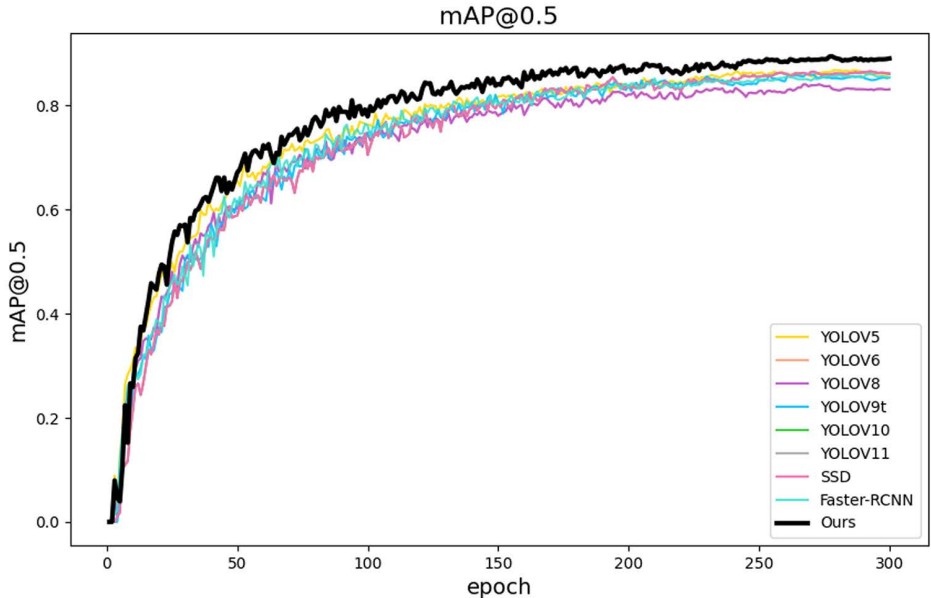

**Fig 9. Comparison of MAP@0.5 results from experiments.**

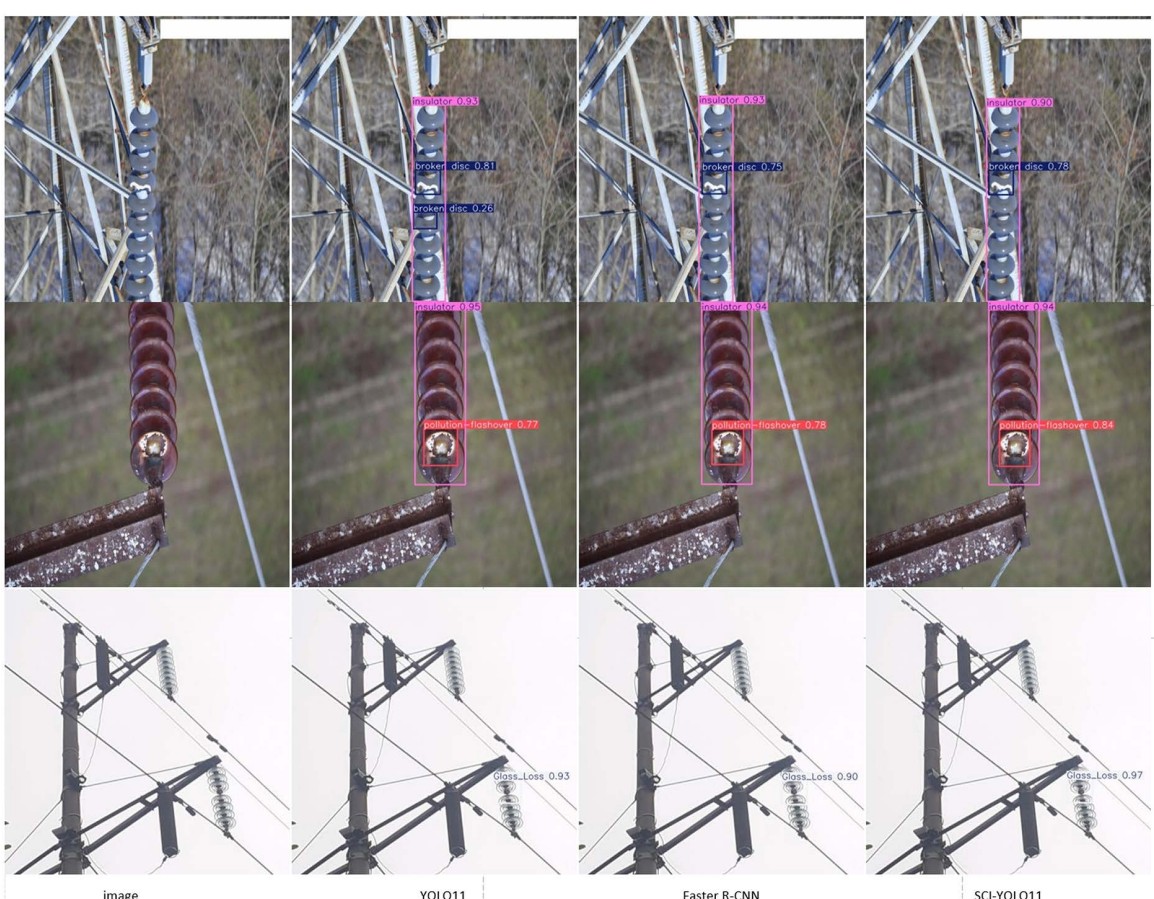

**Fig 10. Visualization of detection results.**

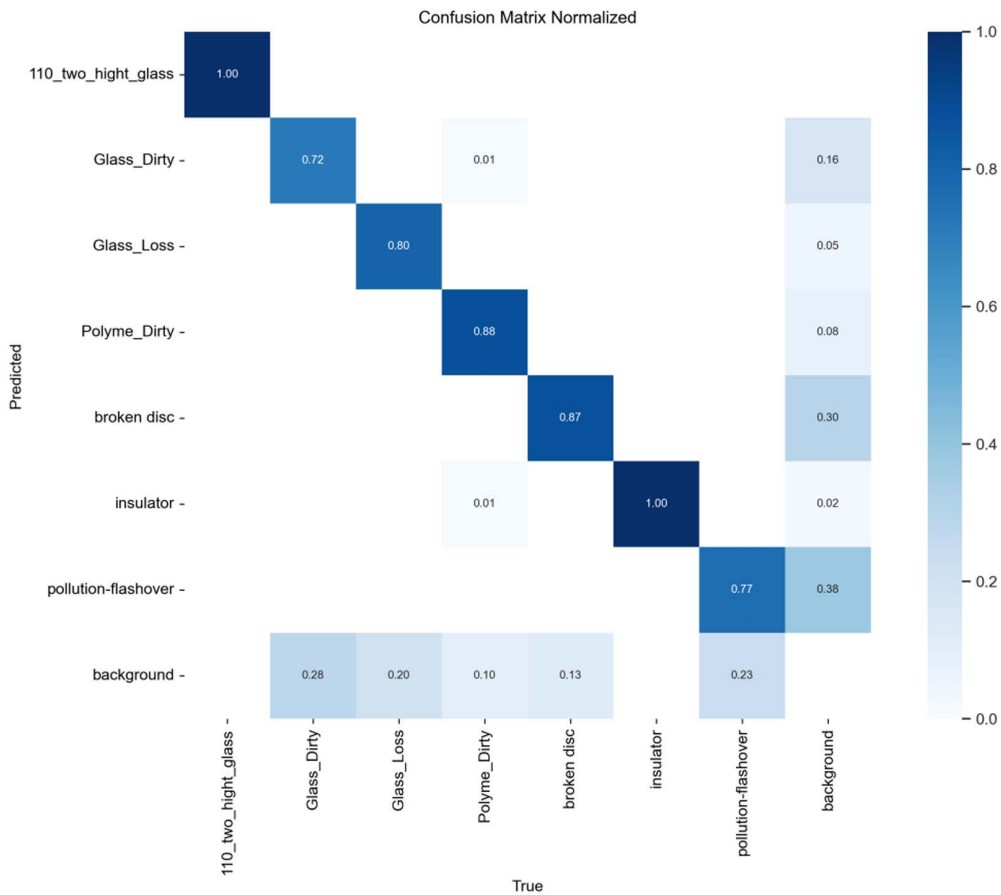

under ideal experimental conditions but also possesses strong adaptability to potential annotation noise and variations in sample distribution.

In order to intuitively validate the effectiveness of SCI-YOLO11, this study conducted a visual analysis of the detection results from YOLO11, Faster R-CNN, and SCI-YOLO11 models. The detection outcomes are illustrated in Fig 10. As shown in the figure, both YOLO11 and Faster R-CNN exhibited instances of false positives and missed detections. Among all the detection results, SCI-YOLO11 demonstrated the highest confidence level.

To evaluate the improved algorithm's performance on defect detection for insulators along power transmission lines using SCI-YOLO11 compared to Faster R-CNN, YOLOv8, and YOLO11 was performed. A confusion matrix analysis was conducted on each model's prediction results to quantitatively assess their classification accuracy across different categories. The values along the diagonal of the confusion matrix represent each model's classification precision for various categories; values closer to 1 indicate better classification performance.

 present the confusion matrices for Faster R-CNN, YOLOv8, and YOLO11, respectively, while Fig 14 illustrates the confusion matrix for SCI-YOLO11. By comparing the diagonal values across different categories, it is evident that SCI-YOLO11 demonstrates superior classification accuracy in most categories. Notably, its performance in detecting Glass-Dirty, Polyme-Dirty, and Broken-Disc targets is particularly remarkable; the diagonal values reached 0.77, 0.92, and 0.89 respectively—significantly outperforming Faster R-CNN (0.72, 0.88, 0.87), YOLOv8 (0.70, 0.89, 0.81), and YOLO11 (0.74,

**Fig 11. Faster-RCNN.**

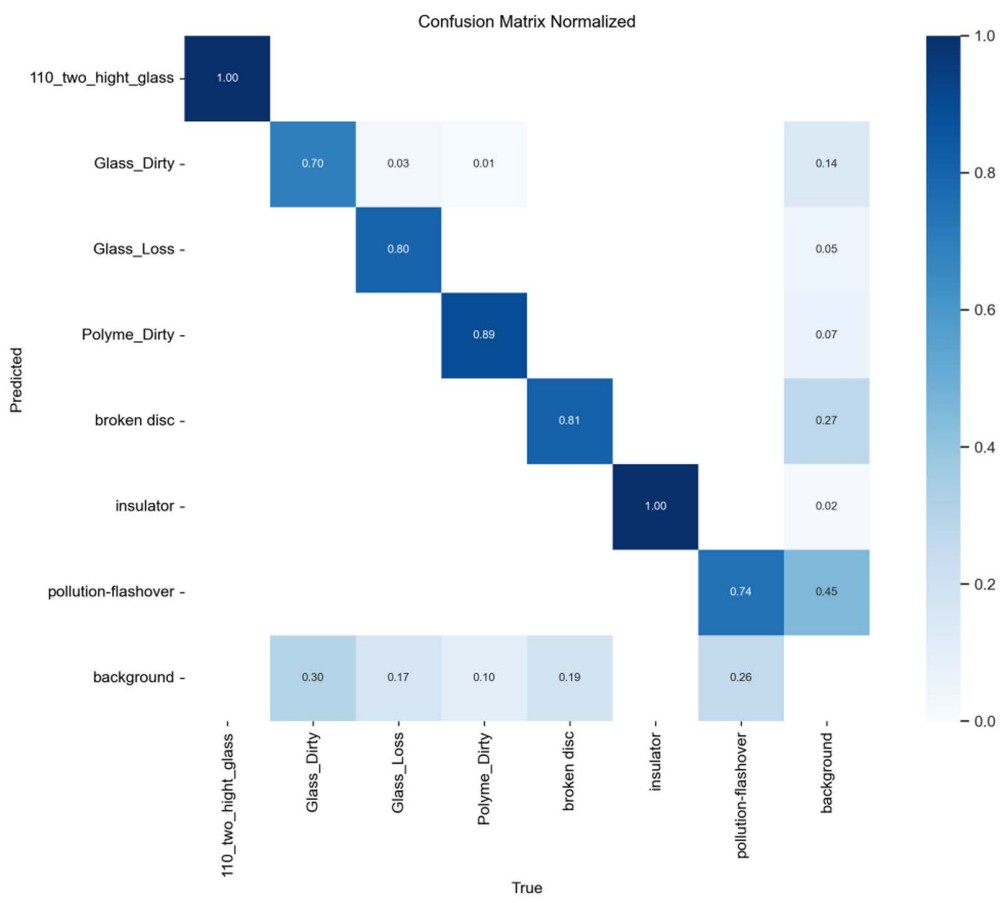

**Fig 12. YOLOV8.**

0.85, 0.84).Among all models evaluated, SCI-YOLO11 exhibits the most pronounced detection capabilities for Glass-Dirty and Broken-Disc categories; this reflects a stronger resilience to interference as well as an advantage in detecting small objects. This enhanced performance can be attributed primarily to the SPDConv module's ability to capture features of small targets effectively; additionally supported by the feature enhancement effects of the SE attention mechanism and optimization provided by Wise-IoU-V3 loss function during bounding box regression.

Furthermore,SCI-YOLO11 outperforms comparative models in terms of average classification accuracy across all categories,indicating that the proposed improvements exhibit greater robustness and stability when addressing challenges posed by complex backgrounds and inconsistent annotation quality.These results further validate SCI-YOLO11's comprehensive performance advantages in object detection tasks,particularly highlighting significant advancements in small object detection as well as suppression of complex backgrounds.

## Conclusion

The present study addresses the demands for efficiency and accuracy in detecting defects in insulators of transmission lines by proposing an improved object detection algorithm-SCI-YOLO11. Compared to traditional YOLO models, SCI-YOLO11 enhances the robustness of feature extraction and the fitting precision of bounding boxes through three key improvements: SPDConv,SE attention mechanism, and Wise-IoU-V3 loss function. Experimental results indicate

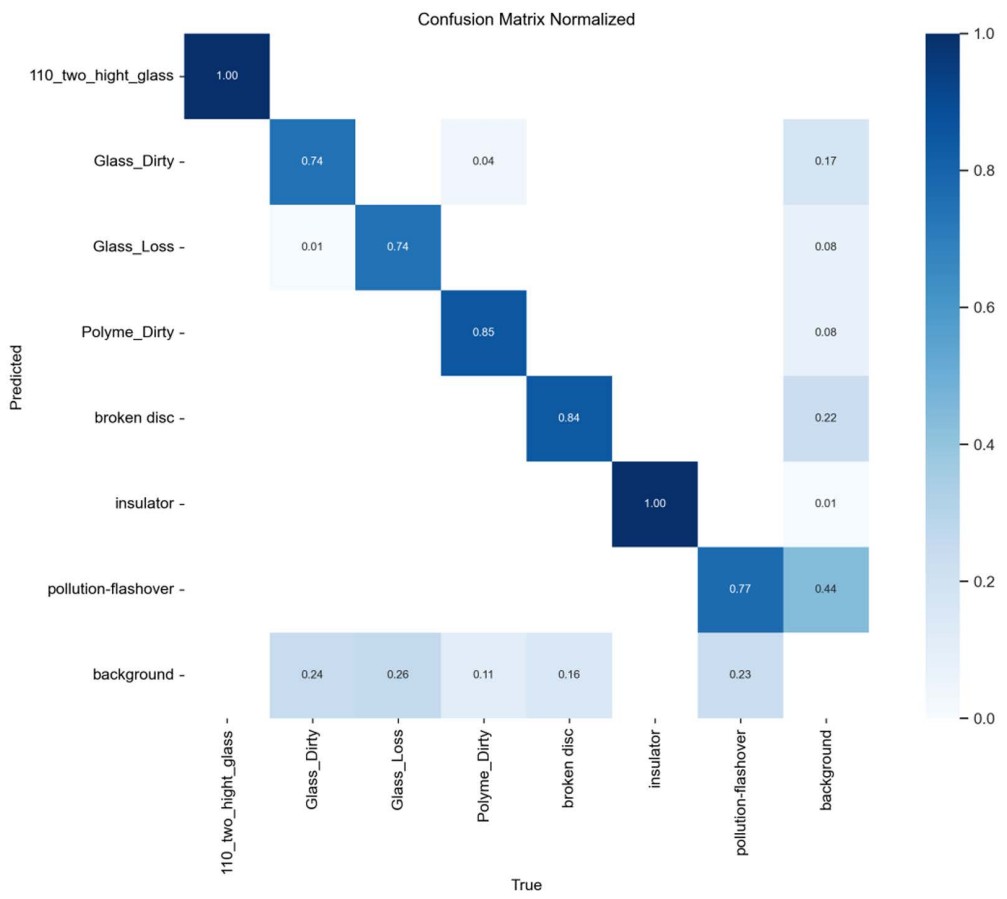

**Fig 13. YOLO11.**

that SCI-YOLO achieves a 3.2% increase in mAP@0.5 over baseline models, reaching an F1 score of 0.87. Furthermore, when compared with various mainstream detection algorithms such as Faster-RCNN, SSD, YOLOv10, YOLOv9t, YOLOv8, YOLOv6, and YOLOv5, SCI-YOLO demonstrates significant advantages in terms of detection accuracy, feature response quality, and control over false positives and missed detections.

Additionally, the visualization results from heatmaps further illustrate SCI-YOLO's superior performance with small targets in complex scenarios. In comparison to baseline models, SCI-YOLO exhibits more concentrated feature responses within target areas while significantly reducing instances of false positives and missed detections. The introduction of SPDConv strengthens the model's ability to express features related to small targets; the SE attention mechanism optimizes weight distribution among features; while the Wise-IoU-V3 loss function improves bounding box fitting effectiveness-resulting in a favorable balance between accuracy, speed, and computational load.

Despite achieving notable research outcomes presented herein, certain limitations warrant further investigation. In our subsequent research endeavors, we plan to explore further possibilities for model lightweighting and precision improvement through advanced optimization methods, including neural architecture refinement and adaptive inference mechanisms, to enhance detection performance. Additionally, we aim to investigate the integration of environmental simulation effects, such as foggy weather conditions, to evaluate and improve the robustness of the model in adverse scenarios. These efforts will lay a solid foundation for advancing insulator defect detection methods and support the safe and stable operation of power transmission systems.

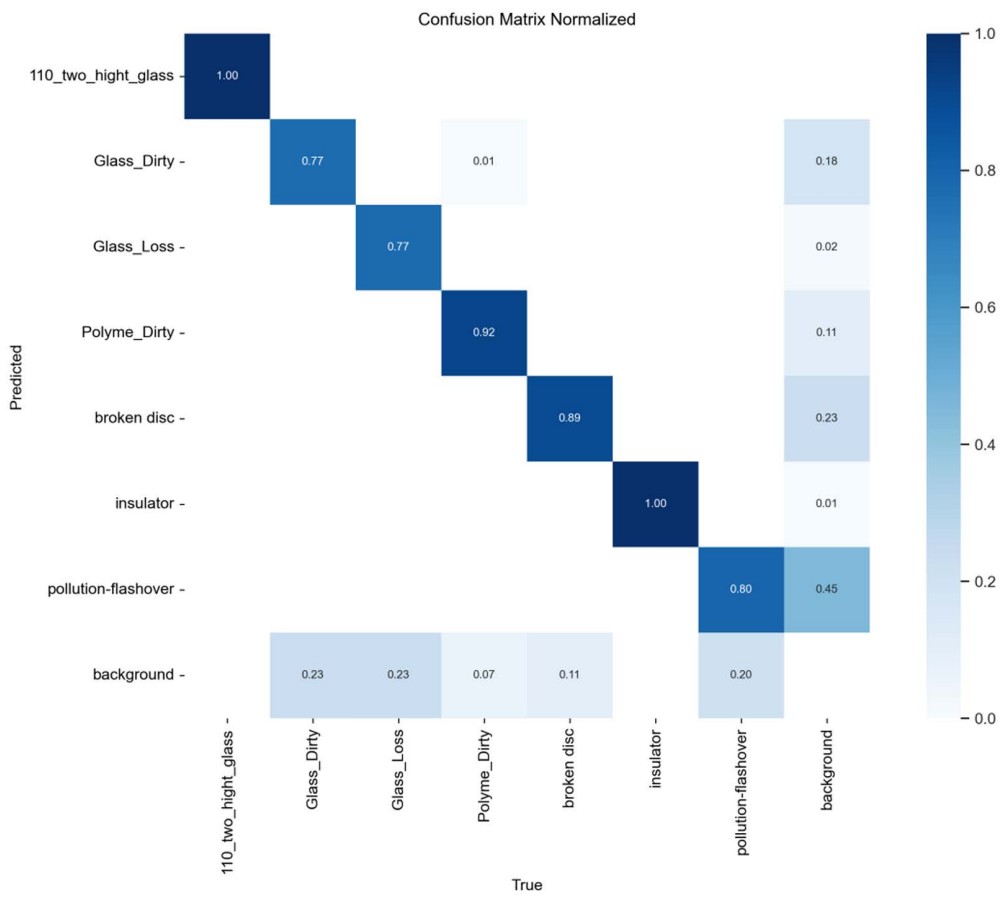

**Fig 14. SCI-YOLO11.**

## Author contributions

**Conceptualization:** Junyan Wang.

**Formal analysis:** Xun Li.

**Investigation:** Miaomiao Wang.

**Methodology:** Junyan Wang, Yuqian Wang.

**Resources:** Baoxi Yuan.

**Software:** Junyan Wang.

**Writing – original draft:** Junyan Wang.

**Writing – review & editing:** Junyan Wang.

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
