## [Decision Letter · Decision Letter 0]

20 Feb 2025

Dear Dr. Wang,

Thank you for submitting your manuscript to PLOS ONE. After careful consideration, we feel that it has merit but does not fully meet PLOS ONE’s publication criteria as it currently stands. Therefore, we invite you to submit a revised version of the manuscript that addresses the points raised during the review process.

We look forward to receiving your revised manuscript.

Kind regards,

Venkatachalam Kandasamy

Academic Editor

PLOS ONE

4. For studies involving third-party data, we encourage authors to share any data specific to their analyses that they can legally distribute. PLOS recognizes, however, that authors may be using third-party data they do not have the rights to share. When third-party data cannot be publicly shared, authors must provide all information necessary for interested researchers to apply to gain access to the data. (https://journals.plos.org/plosone/s/data-availability#loc-acceptable-data-access-restrictions)

4) All necessary contact information others would need to apply to gain access to the data.

6. Please ensure that you refer to Figures 11, 12, 13, 14 in your text as, if accepted, production will need this reference to link the reader to the figures.

7. Please include a caption for figure 5.

8. We note you have included a table to which you do not refer in the text of your manuscript. Please ensure that you refer to Table 1, 2 in your text; if accepted, production will need this reference to link the reader to the Table.

Additional Editor Comments:

Based on the reviewer’s comments, please revise the article as thoroughly as possible.

Reviewers' comments:

Reviewer's Responses to Questions

**Comments to the Author**

1. Is the manuscript technically sound, and do the data support the conclusions?

Reviewer #1: Yes

Reviewer #2: Yes

Reviewer #3: Yes

2. Has the statistical analysis been performed appropriately and rigorously?

Reviewer #1: Yes

Reviewer #2: Yes

Reviewer #3: N/A

3. Have the authors made all data underlying the findings in their manuscript fully available?

Reviewer #1: Yes

Reviewer #2: Yes

Reviewer #3: Yes

4. Is the manuscript presented in an intelligible fashion and written in standard English?

Reviewer #1: Yes

Reviewer #2: Yes

Reviewer #3: Yes

Reviewer #1: 1. Check for grammatical, punctuation, and spelling errors.

2. Use more significant 4 to 6 keywords.

3. Use the references correctly. Some in-text citations are missing.

4. Add a summary table of previous works in related work section.

5. Figure 2 is not clear and readable.

6. Improve figure 3 quality.

7. Use updated references. You can cite this paper >>

M. R. Islam et al., "Crime Prediction by Detecting Violent Objects and Activity Using Pre-Trained YOLOv8n and MoViNet A0 Models," 2023 International Conference on Modeling & E-Information Research, Artificial Learning and Digital Applications (ICMERALDA), Karawang, Indonesia, 2023, pp. 44-49, doi: 10.1109/ICMERALDA60125.2023.10458156.

Reviewer #2: 1- What are the researcher's contributions? Please explain them in brief points.

2- The working steps of the proposed algorithm are not explained algorithmically.

3- Include a link to the dataset used in the reference list.

4- Clarify future work well for this work and all aspects

5- Include additional references to research published between 2022-2024

Reviewer #3: 1. Correct all formatting issues, including "Error! Reference source not found."

2. Expand the explanation of the Wise-IoU-V3 loss function with a clearer derivation.

3. Improve the related work section by explicitly differentiating the proposed method from existing approaches.

4. Provide more details on the dataset’s diversity and representativeness.

**Do you want your identity to be public for this peer review?** For information about this choice, including consent withdrawal, please see our Privacy Policy

Reviewer #1: No

Reviewer #2: No

Reviewer #3: No

---

## [Author Response · Author response to Decision Letter 1]

6 Mar 2025

Responeses to Editor

Reply Thank you very much for your valuable feedback. I have carefully reviewed the PLOS ONE style requirements and made the necessary revisions to ensure compliance. Specifically, I optimized the formatting of the author information to align with the guidelines, corrected improper punctuation usage (e.g., adding spaces after punctuation marks to improve readability), adjusted the manuscript’s line spacing to double spacing as required, and removed numbering from section titles, as it is not included in the official template. To make the revisions clear and transparent, the changes in the manuscript have been highlighted, with the original text marked in yellow and the revised content shown in green. Additionally, I conducted a thorough review of the entire manuscript to address any spelling errors, punctuation mistakes, or other formatting inconsistencies. I sincerely hope these adjustments meet the journal’s requirements and greatly appreciate your guidance and support.

Junyan Wang1, Yuqian Wang2*,Xun Li1, Baoxi Yuan3,Miaomiao Wang4

1 Xijing University, Xi’an, Shaanxi,710123, china

2Graduate Office, Xijing University, Xi'an, Shaanxi 710123, China

3Xi’an Key Laboratory of High Precision Industrial Intelligent Vision Measurement Technology, Xijing University, Xi’an, Shaanxi, China,

4North China Electric Power University, Baoding, 071051, china

* sunbaby0830@sohu.com

Reply Thank you for bringing this to my attention. In response to your comment, I have ensured that the author-generated code is shared in compliance with PLOS ONE's guidelines for code sharing. The code has been uploaded to GitHub at the following link: https://github.com/Xiaobaisci/SCI-yolo11.git. This information has also been included in the manuscript’s abstract to facilitate reproducibility and reuse. I hope this meets the journal’s requirements, and I appreciate your guidance on this matter.

Our code has been uploaded to: https://github.com/Xiaobaisci/SCI-yolo11.git.

Reply Thank you for the suggestion regarding language usage, spelling, and grammar. I have carefully reviewed the manuscript to ensure it meets the required standards. For example, in the sentence "Complex Background Interference Issue: Images captured during insulator inspections frequently contain intricate backgrounds including wires trees along with weather variations that hinder effective recognition of targets," I revised "wires trees" to "wires, trees" to clearly express the list and avoid ambiguity. Additionally, I corrected instances where non-English quotation marks (e.g., “’ ”) were mistakenly used and replaced them with proper English punctuation. I hope these efforts have improved the clarity and accuracy of the manuscript.

4. For studies involving third-party data, we encourage authors to share any data specific to their analyses that they can legally distribute. PLOS recognizes, however, that authors may be using third-party data they do not have the rights to share. When third-party data cannot be publicly shared, authors must provide all information necessary for interested researchers to apply to gain access to the data. (https://journals.plos.org/plosone/s/data-availability#loc-acceptable-data-access-restrictions)

4) All necessary contact information others would need to apply to gain access to the data.

Reply Thank you for your comments regarding data availability. The dataset used in this study is not third-party data but rather our proprietary dataset. It has been uploaded to the Kaggle platform for researchers to access and use.

The dataset can be accessed at the following link: [https://www.kaggle.com/datasets/jmwangdeep/junyan-wang-insulator-dataset].

We hope that the availability of this dataset will contribute to research reproducibility and transparency. Please feel free to contact us if further clarification is needed.

This dataset has been made publicly available on Kaggle (https://www.kaggle.com/datasets/jmwangdeep/junyan-wang-insulator-dataset) to support further research in related fields.

Reply Thank you for the reminder regarding the ORCID iD requirement for the corresponding author. I have ensured that my ORCID iD (0009-0008-8664-2720) is correctly entered and validated in Editorial Manager by following the provided instructions. Please let me know if any further action is needed.0009-0008-8664-2720

6. Please ensure that you refer to Figures 11, 12, 13, 14 in your text as, if accepted, production will need this reference to link the reader to the figures.

Reply�I sincerely apologize for the oversight regarding the references to Figures 11, 12, 13, and 14 in the text. I have now carefully reviewed the manuscript, ensured that these figures are properly captioned, and added the necessary references to them in the text. Additionally, I have standardized all figure references throughout the manuscript to maintain consistency and clarity. Thank you for bringing this to my attention, and I appreciate your understanding.

7. Please include a caption for figure 5.

Reply Thank you for the reminder. I have added an appropriate caption for Figure 5 in the manuscript. Please let me know if any further adjustments are needed.

Fig5. The Process of Data Collection and Annotation by Drones

8. We note you have included a table to which you do not refer in the text of your manuscript. Please ensure that you refer to Table 1, 2 in your text; if accepted, production will need this reference to link the reader to the Table.

Reply Thank you very much for pointing out this issue. I sincerely apologize for the oversight regarding the missing references to Table 1 and Table 2 in the text. I have carefully reviewed the manuscript, ensured that appropriate captions have been added to both tables, and included the necessary references to them in the text. I greatly appreciate your attention to detail and your kind reminder, and I hope the revisions now meet the journal’s requirements.

Reviewer #1: Comment1. Check for grammatical, punctuation, and spelling errors.

1�Reply Thank you for the valuable suggestion. I have carefully reviewed the manuscript for grammatical, punctuation, and spelling errors. Specifically, I corrected instances where non-English punctuation marks such as “’” were mistakenly used and replaced them with proper English punctuation. Additionally, I ensured that spaces are included after periods and other punctuation marks to improve readability. For example, in the sentence "Images captured during insulator inspections frequently contain intricate backgrounds including wires trees along with weather variations that hinder effective recognition of targets," I revised "wires trees" to "wires, trees" to clearly express the list and avoid ambiguity. I hope these revisions address the concerns and enhance the manuscript’s clarity and accuracy.

2. Comment:Use more significant 4 to 6 keywords.

2.Reply:Thank you for the valuable feedback regarding the keywords. We have updated them to better reflect the focus of our work. The revised keywords are: Insulator Defect Detection, YOLO11 Improvement, Small Object Detection, Lightweight Design. We hope these provide a more precise representation of our study's contributions.

3. Use the references correctly. Some in-text citations are missing.

3.Reply:Thank you for pointing out the issue with missing in-text citations. I sincerely apologize for the oversight. For example, in the sentence "Among these challenges lies the issue of detecting small targets like insulator defects; their characteristics may be lost within deep convolutional networks (CNNs), leading to missed detections or false positives [3]," the reference [3] was initially missing but has now been corrected. Additionally, I conducted a thorough review of all references in the manuscript, updated missing citations, and ensured their proper usage throughout the text. I hope these revisions address the concern, and I greatly appreciate your attention to this matter.

Among these challenges lies the issue of detecting small targets like insulator defects; their characteristics may be lost within deep convolutional networks (CNNs), leading to missed detections or false positives [3].

The rapid development of deep learning has provided new technological solutions to address these issues. For example, crime prediction research has successfully employed YOLOv8n for detecting small and intricate targets like weapons, achieving robust results even in complex scenarios [3].

Liu, C., Wu, Y., Liu, J., & Han, J. (2021). MTI-YOLO: A light-weight and real-time deep neural network for insulator detection in complex aerial images. Energies, 14(5), 1426.

4.Comment: Add a summary table of previous works in related work section.

4.Reply:Thank you for your constructive suggestion. Following your advice, we have added a summary table of previous works in the related work section to provide a clearer and more comprehensive comparison of methodologies, key improvements, and limitations in insulator defect detection research. This table helps highlight the contributions and innovations of our proposed SCI-YOLO11 algorithm. We hope this enhancement meets your expectations.

5.Comment: Figure 2 is not clear and readable.

5.Reply:Thank you for your valuable feedback. We have redrawn Figure 2 using Visio to ensure better clarity and readability. We hope the updated version meets your expectations.

6. Comment:Improve figure 3 quality.

6.Reply:Thank you for your valuable feedback. We have redrawn Figure 3 using Visio to ensure better clarity and readability. Additionally, we have included computational details in the "Attention Mechanism: SE Attention" section to enhance its informativeness. We hope the updated figure meets your expectations.

7.Comment: Use updated references. You can cite this paper >>

M. R. Islam et al., "Crime Prediction by Detecting Violent Objects and Activity Using Pre-Trained YOLOv8n and MoViNet A0 Models," 2023 International Conference on Modeling & E-Information Research, Artificial Learning and Digital Applications (ICMERALDA), Karawang, Indonesia, 2023, pp. 44-49, doi: 10.1109/ICMERALDA60125.2023.10458156.

7.Reply:Thank you for your valuable suggestion. I have updated the references in my manuscript to ensure accuracy and relevance.

Additionally, I have incorporated the recommended citation:

M. R. Islam et al., "Crime Prediction by Detecting Violent Objects and Activity Using Pre-Trained YOLOv8n and MoViNet A0 Models," 2023 International Conference on Modeling & E-Information Research, Artificial Learning and Digital Applications (ICMERALDA), Karawang, Indonesia, 2023, pp. 44-49, doi: 10.1109/ICMERALDA60125.2023.10458156.

The findings in this paper demonstrate the adaptability and effectiveness of deep learning models in small object detection, offering valuable insights for insulator defect recognition.

For example, crime prediction research has successfully employed YOLOv8n for detecting small and intricate targets like weapons, achieving robust results even in complex scenarios [3]. These findings demonstrate the adaptability and effectiveness of deep learning models for small-object detection, providing valuable insights for insulator defect identification.

Reviewer #2: 1- What are the researcher's contributions? Please explain them in brief points.

Reply:Thank you for your valuable feedback!

The main contributions of this study are as follows:

Dataset Construction: Constructed and publicly released a diverse insulator defect detection dataset to address the limitations of existing datasets and support further research in the field;

Model Architecture Optimization: Introduced the SPDConv module to optimize feature extraction, significantly improving small target detection accuracy while reducing model parameters;

Attention Mechanism Enhancement: Integrated the SE attention mechanism to enhance feature representation, significantly improving detection performance in complex backgrounds;

Loss Function Improvement: Introduced the WIoU loss function to optimize the bounding box regression process, enhancing the model's adaptability to low-quality samples and overall robustness.

2- The working steps of the proposed algorithm are not explained algorithmically.

Reply:Thank you for your valuable comment regarding the lack of algorithmic explanation of the working steps of the proposed method.

In response, we have provided detailed algorithmic descriptions of the principles behind SPDConv, SE Attention Mechanism, and Loss Functions. The explanations include step-by-step derivations, relevant formulas, and tabular representations to ensure clarity and precision. These updates are included in the Response to Reviewers document for better visibility and understanding.

We appreciate your feedback, which has significantly contributed to improving the scientific rigor and comprehensibility of our work.

3- Include a link to the dataset used in the reference list.

Reply:Thank you for your thorough review and valuable suggestion! We completely agree with your recommendation to include the dataset link explicitly in the reference section. While we previously provided the dataset link in the Related Work section of the manuscript, your suggestion to add it to the references is more formal and enhances the scientific rigor and reproducibility of the work. In the revised manuscript, we have followed your advice and adde

---

## [Decision Letter · Decision Letter 1]

24 Mar 2025

SCI-YOLO11: An Improved Defect Detection Algorithm for Transmission Line Insulators Based on YOLO11

PONE-D-24-60268R1

Dear Dr. Wang,

We’re pleased to inform you that your manuscript has been judged scientifically suitable for publication and will be formally accepted for publication once it meets all outstanding technical requirements.

Kind regards,

Venkatachalam Kandasamy

Academic Editor

PLOS ONE

Additional Editor Comments (optional):

As per reviewers suggestions, the paper has been accepted for publication.

Reviewers' comments:

Reviewer's Responses to Questions

**Comments to the Author**

Reviewer #1: All comments have been addressed

Reviewer #2: (No Response)

Reviewer #3: All comments have been addressed

2. Is the manuscript technically sound, and do the data support the conclusions?

Reviewer #1: Yes

Reviewer #2: (No Response)

Reviewer #3: Yes

3. Has the statistical analysis been performed appropriately and rigorously?

Reviewer #1: Yes

Reviewer #2: (No Response)

Reviewer #3: N/A

4. Have the authors made all data underlying the findings in their manuscript fully available?

Reviewer #1: Yes

Reviewer #2: (No Response)

Reviewer #3: Yes

5. Is the manuscript presented in an intelligible fashion and written in standard English?

Reviewer #1: Yes

Reviewer #2: (No Response)

Reviewer #3: Yes

Reviewer #1: Thanks to the authors for addressing the comments I have mentioned in the earlier review. The manuscript has been improved.

Reviewer #2: (No Response)

Reviewer #3: Thank you for revising the paper. All my comments has been addressed satisfactorily. I have no further queries.

**Do you want your identity to be public for this peer review?** For information about this choice, including consent withdrawal, please see our Privacy Policy

Reviewer #1: No

Reviewer #2: No

Reviewer #3: No

---

## [Editor Report · Acceptance letter]

PONE-D-24-60268R1

PLOS ONE

Dear Dr. Wang,

I'm pleased to inform you that your manuscript has been deemed suitable for publication in PLOS ONE. Congratulations! Your manuscript is now being handed over to our production team.

Kind regards,

on behalf of

Dr. Venkatachalam Kandasamy

Academic Editor

PLOS ONE